# Males of *Dalbulus maidis* Attract Females Through Volatile Compounds with Potential Pheromone Function: A Tool for Pest Management

**DOI:** 10.3390/insects16101021

**Published:** 2025-10-02

**Authors:** Mateus Souza Sanches, Miguel Borges, Raul Alberto Laumann, Charles Martins Oliveira, Marina Regina Frizzas, Maria Carolina Blassioli-Moraes

**Affiliations:** 1Laboratório de Semioquímicos, Embrapa Recursos Genéticos e Biotecnologia, Brasília, DF 70770-917, Brazil; miguel.borges@embrapa.br (M.B.); raul.laumann@embrapa.br (R.A.L.); 2Programa de Pós-Graduação em Ecologia, Universidade de Brasília, Brasília, DF 70910-900, Brazil; frizzas@unb.br; 3Embrapa Cerrados, Planaltina, DF 73310-970, Brazil; charles.oliveira@embrapa.br

**Keywords:** semiochemicals, chemical communication, olfactometry, corn leafhopper, pest management, Hemiptera, Cicadellidae

## Abstract

Insects use chemical compounds for communication, and sex pheromone is one of the most important signals used by males and females to find each other for mating purposes. The corn leafhopper, *Dalbulus maidis*, is an insect vector that transmits pathogens causing diseases in maize crops, but it was unknown whether it uses sex pheromones in their communication. In this study, we tested whether *D. maidis* produces volatile compounds that attract the opposite sex. We collected volatiles from live insects and evaluated their influence on the behavioral responses of conspecifics. We found that males produce odors that attract females. Interestingly, males avoided odors emitted by stressed females, which may suggest the release of an alarm pheromone. These findings highlight for the first time the role of semiochemicals in intraspecific *D. maidis* communication, and open new perspectives for the development of monitoring and management tools targeting this important pest.

## 1. Introduction

Insects rely on chemical communication to interact with conspecifics (pheromones) or with organisms from other species (allelochemicals) [1,2]. Sex pheromones are chemical compounds released by sexually mature individuals that function to stimulate the opposite sex for mate location and copulation. These pheromones can also convey information about mate recognition, attraction, reproductive status, and the fitness of the emitter [3].

The corn leafhopper, *Dalbulus maidis* (DeLong & Wolcott), is a phloem-feeding insect that primarily causes indirect damage to maize *Zea mays* L., its main host plant [4], due to its ability to efficiently transmit the pathogens associated with the corn stunt disease complex. These include corn stunt spiroplasma (CSS, *Spiroplasma kunkelii*), maize bushy stunt phytoplasma (MBSP, *‘Candidatus* Phytoplasma asteris’), as well as maize rayado fino virus (MRFV) and maize striate mosaic virus (MSMV) [4,5,6]. Currently, no curative management measures are available for these diseases [7,8].

The mating behavior of *D. maidis* has been previously described, including the use of vibrational and acoustic signals during courtship [9,10]. However, no studies have investigated whether this species emits long-range chemical signals, like sex pheromones, considering that the acoustic signals produced by *D. maidis* are transmitted only over short distances when both sexes are on the same substrate [9]. Chemical communication is, nevertheless, known to play a role in host plant selection by *D. maidis* [11,12].

Although several studies have demonstrated the use of acoustic signals among leafhoppers, for a long time, there was no evidence of molecules functioning as sex pheromones in any species within the group (Hemiptera: Auchenorrhyncha: Cicadomorpha). This scenario began to change about a decade ago with the identification of an aggregation pheromone in *Callitettix versicolor* (Fabricius) [13], and, more recently, the first evidence of sex pheromones was reported for *Philaenus spumarius* (L.), although the chemical structure has not yet been identified [14].

Due to their high specificity and the remarkable sensitivity of insect olfactory systems, sex pheromones have been widely studied as tools for agricultural pest management [15,16]. Since it is a vector insect, there is no established economic threshold for *D. maidis*, and its management still relies primarily on the systematic use of chemical insecticides during the early stages of maize cultivation, which is the critical period for pathogen transmission [12,17]. Therefore, alternative control strategies are urgently needed, including the potential use of sex pheromones for monitoring and management.

In this context, the objective of this study was to investigate whether *D. maidis* emits volatile compounds that can influence the behavioral response of conspecifics, with a particular focus on the presence of a sex pheromone.

## 2. Materials and Methods

### 2.1. Corn Leafhopper-Dalbulus Maidis

*Dalbulus maidis* (DM) used in this study were obtained from a colony established in 2022 at Embrapa Cerrados (Planaltina, DF, Brazil), originally collected from adult individuals in experimental maize fields (15°36′16”S, 47°42′38”W). The colony was subsequently decontaminated following the recommended procedures [18] to eliminate potential carriers of pathogens (mollicutes and viruses), ensuring a healthy colony.

The individuals used in the experiments were randomly selected from adult insects, without controlling for age, reproductive status, or mating history. To determine the sex, insects were placed in glass tubes and examined under a stereomicroscope. Females were identified by the presence of an ovipositor at the tip of the abdomen (Appendix A).

### 2.2. Maize

Maize plants of the Synthetic Spodoptera (SS) genotype were used in the experiments since their herbivore-induced plant volatiles are well known [19]. The seeds were obtained from the germplasm bank of Embrapa Maize and Sorghum (Sete Lagoas, MG, Brazil) and were sown in plastic pots (0.3 L) with a mixture of natural soil (Latosol) and organic substrate (Max Fertil, Santa Catarina, Brazil, composition: pine bark, natural phosphate, carbonized rice husk, vermiculite, chemical fertilizer NPK) in a proportion of 1:1 w/w, without post-fertilization. Plants were kept in a greenhouse (Brasília, DF, Brazil) under natural conditions of temperature, humidity, and photoperiod (14L/10D; Brasília, DF, 15°46′46″ S and 47°55′46 W) and manually irrigated with a watering can every two days. For experiments, plants were used at the V2 stage (two expanded leaves, 4 to 7 days after germination).

### 2.3. Volatile Collection

Volatile organic compounds (VOCs) emitted by *D. maidis* were collected using dynamic headspace aeration systems across two experimental sets that lasted for 2 weeks each. In the first experimental set three treatments were established: DM: 100 unsexed adult leafhoppers; DM-Maize: 100 unsexed adults plus one maize plant at V2 growth stage, and Maize: a maize plant at the V2 stage. The objectives were as follows: DM to assess volatiles emitted exclusively by the leafhoppers; DM-Maize to evaluate whether volatiles are only emitted in the presence of a food resource; Maize to identify and exclude volatiles released by the plant itself. For each treatment 20 replicates were conducted. The maize plants were placed intact in the system, with aluminum covering the soil to prevent volatiles from originating from the soil or roots (Appendix A).

In the second experimental set, two other treatments were conducted: volatiles were collected from 100 males or 100 females and another with one maize plant at the V2 growth stage, these treatments were called: DM-Male (100 males) and DM-Female (100 females). For each treatment 10 replicates were conducted.

All treatments were conditioned in glass chambers (2L volume; 14 cm × 24 cm) connected to a system supplied with charcoal-filtered air (4–20 mesh, Supelco, PA, USA) at a constant flow rate of 1.0 L min^−1^. Simultaneously, a vacuum pump (LGI-DVP-1, ultimate vacuum 200 mbar, pumping speed 60 L/min, LGI Scientific, São Paulo, Brazil) maintained an outflow of 0.8 L min^−1^, ensuring continuous air circulation through the chamber and directing volatiles into an adsorbent tube filled with HayeSep Q (polydivinylbenzene copolymer, 100 mg, 80–100 mesh, Supelco, Pennsylvannia, USA).

Volatiles trapped on adsorbent filters were eluted every 24 h with n-hexane, with 72 h accumulations during weekends. All insects and plants were replaced weekly with new ones. For the DM treatment, which did not include maize plants, six replicates were conducted, and with volatiles collected for 24 h, due to high mortality (individuals typically died after 24 h without food).

### 2.4. Chemical Analysis

All volatile samples were analyzed using an Agilent 7890A gas chromatograph equipped with a flame ionization detector (GC-FID) and a non-polar DB-5MS column (30 m × 0.25 mm ID, 0.25 μm film thickness, Supelco, PA, USA). The oven temperature was initially set at 40 °C for 2 min, then ramped at 5 °C min^−1^ until reaching 180 °C, held for 0.1 min, followed by an increase of 10 °C min^−1^ to 250 °C, where it was held for 20 min. The injector was set to 250 °C and the FID to 270 °C. Each sample (2 μL) was injected in splitless mode, with helium as the carrier gas. Chromatographic data acquisition was performed using the GC ChemStation software (Agilent, Santa Clara, CA, USA, version 2.4). For qualitative analyses selected volatile samples was analyzed using an Agilent 5975 mass selective detector (GC-MS) (Agilent, Santa Clara, CA, USA) coupled with a quadrupole mass analyzer, equipped with the same non-polar DB-5MS capillary column (30 m × 0.25 mm ID, 0.25 μm film thickness; J&W Scientific, Folsom, CA, USA). Injections were performed in splitless mode with 2 μL of sample, and helium was used as the carrier gas. Ionization was conducted by electron impact (EI) at 70 eV, with the ion source temperature set at 230 °C. The oven program followed the same temperature profile as the GC-FID analyses: 40 °C for 2 min, ramped at 5 °C min^−1^ to 180 °C, followed by 10 °C min^−1^ to 250 °C, with a final hold of 20 min. Data acquisition and analysis were conducted using the MassHunter Qualitative software (version 10.1, Agilent, Santa Clara, CA, USA). Compound identification was performed by comparing the mass spectra to those in the NIST library [20] and to published spectra, along with retention index calculations based on the DB-5MS column. Tentative identifications were confirmed by co-injection with authentic standards, either commercially sourced or synthesized in-house.

### 2.5. Chemicals

Authentic chemical standards of decane (99%), tetradecane (99%), pentadecane and hexadecane (99), octanal (≥98.0% (GC)), nonanal (98%), decanal (98%), dodecanal (97%), camphene (95%), α-pinene (98%), β-pinene (99%), 3-carene (≥90%), limonene (97%), methyl salicylate (99%), 6-methyl-5-hepten-2-one (99%), β-caryophyllene (98%), geranylacetone (97%) and cyclosativene (99%) were purchased from Sigma-Aldrich (Steinheim, Germany). Linalool was purchased from TCI America (Portland, USA). (*E*)-4,8-Dimethylnona-1,3,7-triene (DMNT) (95%) was synthesized from geraniol [21]. The solvent hexane (97%) was purchased from Sigma-Aldrich (Steinheim, Germany) and redistilled before use.

### 2.6. Olfactometer Bioassays

To assess whether *D. maidis* is attracted to the opposite sex through chemical cues, olfactometer bioassays were conducted using a Y-shaped dual-choice olfactometer (19 cm × 19 cm; choice arms: 7 cm × 1.7 cm; main arm: 5 cm × 1.7 cm, Acrilico Arte, Brasília, DF, Brazil). Before the experiments with the odor sources, the behavioral response of insects of both sexes were evaluated when exposed to clean air to assess any potential bias that might affect their choice. Then, female and male responses were tested separately by contrasting the odor of 20 live males or 20 live females and clean air in the following pairs: air–air, male–female, male–air, and female–air. Odor sources were generated by placing the insects inside a 40 mL glass container (5 cm × 6 cm) without food. The insects were transferred to the experimental arenas using an insect mouth aspirator [18], from the rearing cages to glass tubes, and, subsequently, to the olfactometers.

Initially, insects were placed into the containers and bioassays were immediately started in the afternoon (n = 30). One insect (male or female) was released at the entrance of the olfactometer, and its behavior was monitored for 10 min. Because the odor source insects in the glass container were highly mobile, bioassays were repeated using acclimated individuals. Acclimation was achieved by keeping the insects intended as odor sources inside the container for two hours. After this period, the odor sources were connected to the olfactometer, resulting in clearer response behaviors from the test insects toward the odors being evaluated.

Since only females had been attracted to the odor of live males in the previous experiment, they were tested with a mixture (10 females and 10 males together) versus males, as well as with male aeration extracts (obtained from 24 h volatile collections, DM-Male) against n-hexane. For male aeration extract bioassays, 5 µL of DM-Male samples (equivalent to the aeration of 20 individuals) were applied to a paper filter (0.5 cm × 0.5 cm) and compared to 5 µL of n-hexane. The paper filters were placed in glass containers (20 mL; 3 cm × 3 cm) and connected to the system. After three bioassays (10 min each), the filter papers containing the 5 µL of the sample, volatile or hexane, were replaced with new ones.

Thirty different individuals of the tested sex were used to evaluate their choice in response to the odor treatments. The parameters measured were first choice (the first arm entered for more than 30 s) and residence time (total time spent in each arm). Individuals that made no choice within this period were classified as non-responsive and excluded until 30 bioassays with a choice were obtained. To avoid contamination with chemical cues, the olfactometer was cleaned after every five bioassays, and odor source positions were alternated to prevent side bias. A maximum of 10 bioassays were conducted per day to minimize the influence of environmental factors such as temperature, humidity, atmospheric pressure and insect condition. All bioassays were performed between 14:00 and 17:00 h.

### 2.7. Statistical Analyses

Olfactometers bioassays data were analyzed using a chi-square test for first choice and paired t-tests for residence time with a 95% level of confidence; all tests were conducted on the R platform [22].

## 3. Results

### 3.1. Bioassays

Insects of both sexes exhibited no directional preference when exposed solely to clean air on both sides in olfactometers (females: χ^2^ = 0.31, *p* = 0.57; *t*-test = 0.41, *p* = 0.2; males: χ^2^ = 0.043, *p* = 0.83; *t*-test = 0.55, *p* = 0.58).

Females discriminated between males and females, being attracted to males (χ^2^ = 6.533, *p* = 0.01; *t*-test = 3.03, *p* = 0.005), however, they did not differentiate males from the mixture of males and females (χ^2^ = 0.533, *p* = 0.46; *t*-test = 1.69, *p* = 0.1) (Figure 1A and B). When exposed to the odor of non-acclimated males, *D. maidis* females did not show a preference for male odor compared to air in terms of first choice (χ^2^ = 3.33, *p* = 0.067), but they spent significantly more time in the arm containing male odor (*t*-test = 25.299, *p* = 0.017) (Figure 1A and B). Using the odor of acclimated males, females corroborated the previous results with non-acclimated males, and showed a clear preference for male odor, both in first choice (χ^2^ = 10.8, *p* = 0.001) and in residence time (*t*-test = 33.104, *p* = 0.002) (Figure 1A and B). Females also showed a preference to male aeration extract compared to hexane (χ^2^ = 8.533, *p* = 0.003; *t*-test = 3.023, *p* = 0.005). Finally, females did not show a significant preference for female odor compared to air (χ^2^ = 0.142, *p* = 0.705; *t*-test = -1.145, *p* = 0.261).

Males exhibited avoidance behavior toward female odor when females were not acclimated for a two-hour period (Figure 2A and B). They showed a significant preference for the arm releasing clean air over the odor of 20 females, both in first choice (χ^2^ = 4.8, *p* = 0.024) and residence time (*t*-test = 22.838, *p* = 0.03). On the other hand, when females were acclimated, males showed no attraction either to female odor or to clean air, displaying no significant preference (χ^2^ = 0.133, *p* = 0.718 for first choice; *t*-test = 0.972, *p* = 0.338 for residence time). They did not discriminate between males and females (χ^2^ = 0.133, *p* = 0.715; *t*-test = –0.09, *p* = 0.924), nor when exposed to male odor, showing no side preference (χ^2^ = 0.615, *p* = 0.432; *t*-test = –0.267, *p* = 0.791).

### 3.2. Volatiles

A total of 30 volatile compounds were identified from headspace collections containing either male or female *D. maidis* individuals that were similar for both sexes (Figure 3). The chemical profile of volatiles obtained when only the insects were used for volatile collection, did not show a qualitative difference between both sexes. The main compounds identified were monoterpenes like α (2) and β (4) pinenes, limonene and linalool (14), sesquiterpenes like β-caryophyllene and aldehydes (Figure 3). The chemical analysis of volatile samples conducted with both sexes did not show differences compared with when the insects had their volatiles collected and separated by sex. When maize plants were used as food sources for the insects during the volatile collections, we did not detect different compounds that could be attributed to the insects. The new peaks observed in the chromatographic profile in the presence of both insects and maize plants were associated with plant-derived compounds, such as DMNT (16) and cyclosativene (26) (Figure 3).

## 4. Discussion

The bioassay results showed that female corn leafhoppers, *D. maidis*, can discriminate between males and females through volatiles, moving toward the odor of live males and male aeration extracts, thereby suggesting that males produce volatiles functioning as a pheromone. Males were not attracted to these conspecific volatiles of both sexes, as the females did not respond to the odor of other females, supporting the hypothesis that these compounds are only released by males and act as sexual attractants exclusively for females. The chemical analysis of volatile samples containing the volatiles emitted by *D. maidis* did not identify male-specific compounds that could explain this attraction. It is possible that the male-specific compounds are produced in very tiny amounts, below the detection limits of the equipment used. While the behavioral assays strongly suggest a role in sexual attraction, the precise function as a sex pheromone remains to be confirmed until the active compound is chemically identified and the behavioral role of the synthetic pheromone is evaluated.

In both conditions used to collect volatiles from males and females, the main compounds identified were aldehydes, monoterpenes, and linear hydrocarbons. No sex-specific compounds were observed between the volatile profiles of males and females collected without food and those of males and females collected with food. In samples with and without food we mainly found compounds related to maize plant emissions, such as cyclosativene and DMNT. The presence of these compounds, even in the absence of maize plants, may be related to honeydew excretion by the leafhoppers, which could not be controlled under our methodology. In *Lycorma delicatula* (White) (Fulgoridae), honeydew volatiles released by males can attract conspecific males [23]; therefore, this may represent a form of chemical communication in Auchenorrhyncha insects.

The absence of food can influence pheromone production, as demonstrated by [24]. The stink bug *Euschistus heros* (Fabricius) ceases sex pheromone production after remaining more than 24 h without food, before this period males of *E. heros* cotinue producing the sex pheromone. In the present study, we were not able to identify sex pheromone compounds and, therefore, cannot determine precisely whether the absence of food affects or not pheromone emission. All insects used in the experiments were fed until the start of the assays, remained without food for only 24 h; thus, we hypothesized that at least during the first hours of the aeration, the insects had physiological conditions to produce and emit semiochemicals.

Male of leafhoppers mate multiple times throughout their lives [9]. Therefore, it is reasonable to expect that mated individuals may continue producing and emitting sex pheromones. The results obtained in this study appear to support this hypothesis, as part of the insects used were mated and males were still able to attract females in olfactometer bioassays, even though females mate only once in their lives [9]. Further research should be conducted using virgin males and females to assess whether virgin individuals exhibit stronger behavioral responses in bioassays and whether they produce higher levels of sex pheromones. However, this attraction, even in mated females, should be considered an advantage for potential control methods in the field.

Mating communication in leafhoppers (Auchenorrhyncha) has been characterized primarily by acoustic and vibrational signals [25,26,27]. However, multiple studies have demonstrated the use of chemical cues in these insects for host-plant location [28,29], including in *D. maidis* [11,12]. Only recently has evidence emerged for the presence of sex pheromones within this suborder. In 2022, the first indication of a sex pheromone was reported for *P. spumarius* (Hemiptera: Auchenorrhyncha: Aphrophoridae), where males were attracted to female odors in Y-tube olfactometer assays [14]. Similarly, *L. delicatula* males were shown to be attracted to female body extracts in Y-tube bioassays [30]. In addition, some studies with *D. maidis* in maize fields using sticky cards collected more males, which was attributed to males being caught in traps while searching for mates and attempting copulation [31,32]; even this cannot be confirmed only by sticky cards. In another study, the escape behavior of *D. maidis* was not affected by sex [33], indicating that the species exhibits variation in different aspects and warrants further investigation to better understand the searching behavior of males and females over both short and long distances. Our results may reflect short-distance attraction mediated by pheromone, since the distance from the odor source to the olfactometer was less than 50 cm.

The mating behavior of *D. maidis* has been previously studied, including sexual behavior and the role of vibrational and acoustic signals [9,10]. Notably, male behaviors such as wing fanning are more pronounced than those of females, which the authors believed might be associated with the release of medium-range chemical signals [9]. This supports our findings in the present study, showing that females are attracted to male odor even in the absence of visual or acoustic cues, demonstrating intraspecific chemical communication in this species. To the best of our knowledge, this is the first study to demonstrate the potential role of volatiles in sexual attraction within the superfamily Membracoidea, suggesting the presence of a pheromone in *D. maidis*.

The results obtained with non-acclimated females could suggest the existence of an alarm pheromone. This is supported by the observation that males preferred clean air over the odor of stressed female conspecifics, indicating avoidance of environments that may signal potential threats [34]. Leafhoppers may use chemical cues to mediate various behaviors, as it is common to observe aggregations of individuals on the same maize plant, similar to aphids, which rely on multiple chemical signals [35]. Likewise, aggregated treehoppers also release alarm pheromones, but only when their body wall is pierced, such as during a predator attack [36].

This study demonstrates that chemical communication plays a role in the reproductive behavior and attraction of *D. maidis*, alongside acoustic signaling. Given that insects must locate mates over long distances, it is reasonable to propose that many members of Auchenorrhyncha may also rely on chemical signals. These findings highlight the need for further investigation into chemical communication in this group.

Sexual pheromones are currently employed in various control strategies [37], including mass trapping with pheromone-baited traps, as is the case for *Tuta absoluta* (Meyrick) [38] mating disruption, which involves the release of synthetic pheromones in the field to reduce mate location and copulation rates [39], population monitoring [40,41,42] and attract-and-kill techniques [43]. Even in the absence of identified sex pheromones, chemical-based control strategies have been developed for other leafhopper species. For instance, a pushh–pull system using plant volatile compounds has been proposed for the tea green leafhopper *Empoasca flavescens* (Cicadellidae) [44,45]. Another system with *L. delicatula* showed effective trap attraction using pheromone lures based on body extracts to capture males and females during the oviposition period [46].

In this way, these findings are a step toward advancing knowledge of the chemical communication of *D. maidis* to understand their biology and ecology. Furthermore, it is also a possibility to develop new strategies to monitor or control this pest using chemical signals or extracts from living insects, specifically to capture females.

## Figures and Tables

**Figure 1 insects-16-01021-f001:**
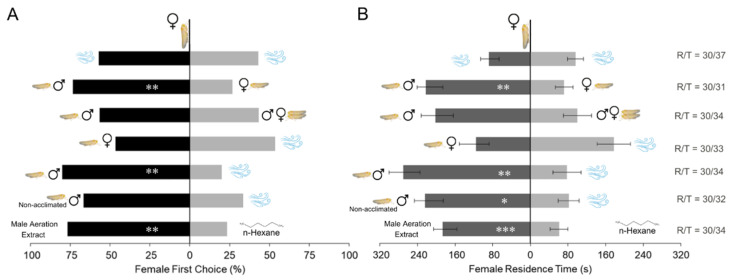
(**A**) First choice of *Dalbulus maidis* females in dual-choice olfactometer bioassays between different odors sources: 20 males, 20 females, 10 males and 10 females, clean air, male aeration extract and n-hexane (n = 30). (**B**) Mean residence time of females during the bioassays. * indicates *p* < 0.05, ** *p* < 0.01 and *** *p* < 0.001 for chi-square tests (A) and t-tests (B). Error bars represent the standard error of the mean. R/T indicates the number of responsive/total number of bioassays conducted.

**Figure 2 insects-16-01021-f002:**
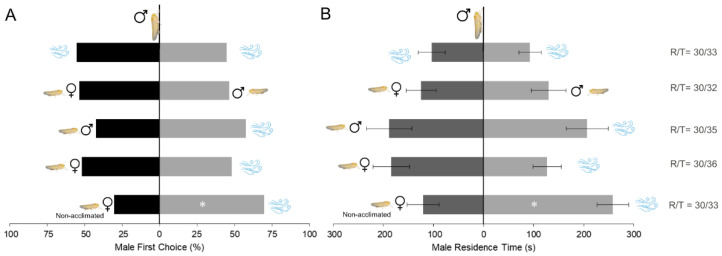
(**A**) First choice of *Dalbulus maidis* males in dual-choice olfactometer bioassays between different odor sources: 20 males, 20 females and clean air (n = 30). (**B**) Mean residence time of males during the bioassays. * indicates *p* < 0.05 for chi-square tests (**A**) and *t*-tests (**B**). Error bars represent the standard error of the mean. R/T indicates the number of responsive/total number of bioassays conducted.

**Figure 3 insects-16-01021-f003:**
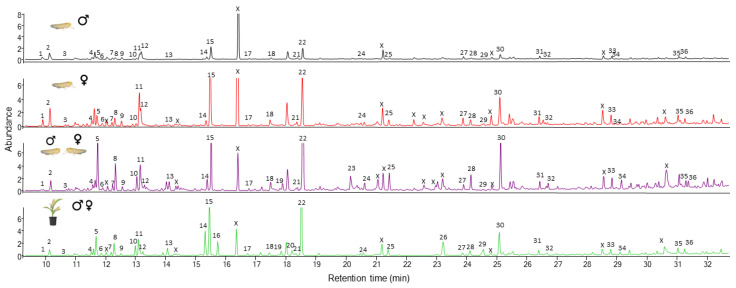
Chromatograms of headspace collections from males, females, both sexes and both sexes with maize plants analyzed by gas chromatography coupled with mass spectrometry (GC-MS). X indicates phthalate group contaminants. 1–β-thujone, 2–α-pinene, 3–camphene, 4–β-pinene, 5–6-methyl-5-hepten-2-one, 6–2-pentylfuran, 7–decane, 8–octanal, 9–3-carene, 10–p-cymene, 11–2-ethyl-1-hexanol, 12–limonene, 13–γ-terpinene, 14–linalool, 15–nonanal, 16–DMNT: (E)-4,8-dimethyl-1,3,7-nonatriene, 17–2-ethylhexyl acetate, 18–NI-1, 19–terpinen-4-ol, 20–methyl salicylate, 21–dodecane, 22–decanal, 23–4-phenyl-2-butanol, 24–NI-2, 25–NI-3, 26–cyclosativene, 27–tetradecane, 28–dodecanal, 29–(E)-β-caryophyllene, 30–geranylacetone, 31–pentadecane, 32–tridecanal, 33–hexadecane, 34–tetradecanal, 35–heptadecane, 36–pentadecanal.. NI = non-identified.

## Data Availability

The raw data supporting the conclusions of this article will be made available by the authors, without undue reservation.

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
