# Peer review of "Males of Dalbulus maidis Attract Females Through Volatile Compounds with Potential Pheromone Function: A Tool for Pest Management"

_insects, 2025, doi:10.3390/insects16101021_

Round 1

Reviewer 1 Report

Comments and Suggestions for Authors

This study focuses on the intraspecific chemical communication mechanism of the corn leafhopper (Dalbulus maidis). It is the first to confirm that volatile compounds released by males of this species are attractive to females, and males avoid the odors of females non-responsive to the environment (which may be related to alarm pheromones). The research fills the gap in the study of sex pheromones in Membracoidea (Hemiptera) and provides a new perspective for pest monitoring and management, with certain scientific value and application potential. The experimental design is basically standardized, and the results support the core conclusions, but some details need to be supplemented and improved.

  1. The term "pheromone" is used in the abstract and keywords, but the results section clearly states that "no sex-specific compounds were detected" (Section 3.3 Results), and the compounds have not been identified. Thus, the choice of this term requires careful consideration.
  2. Materials and Methods2.1:it is stated that "the individuals used in the experiment were randomly selected from adults, with no control over age, reproductive status, or mating history." However, the production and release of sex pheromones are often influenced by these factors. It is recommended that in key bioassays (such as the attraction of females to male volatiles), clearly unmated individuals of similar age should be used, and their reproductive status should be reported.
  3. Materials and Methods2.3: For the DM treatment group (without food), volatile compounds were only collected for 24 hours due to high mortality, whereas other groups (e.g., DM-Male, DM-Female) had longer collection durations (eluted every 24 hours, with 72-hour accumulation on weekends). Hunger stress may significantly alter the volatile emission profiles of insects. It is necessary to emphasize this limitation in the discussion, or design experiments to ensure that insects in all treatment groups are in a similar physiological state.
  4. Materials and Methods2.6: It is mentioned that "individuals that did not make a choice were excluded until 30 valid bioassays were obtained". It is necessary to report the total number of tested individuals and the proportion of excluded (non-responsive) ones, so as to evaluate the representativeness of the experimental results and potential biases.
  5. It is recommended that in Figures 1 and 2, the results of statistically non-significant tests be uniformly labeled "ns".
  6. 3.1 Results:It is described that males avoid the odor of "non-acclimated" females, but show no preference for the odor of "acclimated" females. Similarly, it is necessary to clarify what "non-acclimated" and "acclimated" specifically refer to (the state of odor-source insects? Or the state of test insects?).
  7. 3.3 Results / 4 Discussion:GC-MS results showed that the volatile compounds of both sexes are "similar" and "no qualitative differences between the two sexes were observed". However, bioassays revealed that females are specifically attracted to male volatiles (and not attracted to female volatiles), while males specifically avoid volatiles from stressed females. This strongly suggests the existence of sex-specific and state-specific compounds, which may be present at concentrations below the detection limit of GC-MS or subject to co-elution. This contradiction needs to be explored more thoroughly in the discussion.
  8. It is mentioned that there are indications of sex pheromones inPhilaenus spumariusand that males of Lycorma delicatula (Fulgoridae) are attracted to female extracts. The discussion should conduct a more in-depth comparison of the similarities and differences between D. maidis in this study (where females are attracted to males) and these known cases (typically males being attracted to females), and explore their significance in the chemical ecology of Hemiptera.

Author Response

This study focuses on the intraspecific chemical communication mechanism of the corn leafhopper (Dalbulus maidis). It is the first to confirm that volatile compounds released by males of this species are attractive to females, and males avoid the odors of females non-responsive to the environment (which may be related to alarm pheromones). The research fills the gap in the study of sex pheromones in Membracoidea (Hemiptera) and provides a new perspective for pest monitoring and management, with certain scientific value and application potential. The experimental design is basically standardized, and the results support the core conclusions, but some details need to be supplemented and improved.

  1. The term "pheromone" is used in the abstract and keywords, but the results section clearly states that "no sex-specific compounds were detected" (Section 3.3 Results), and the compounds have not been identified. Thus, the choice of this term requires careful consideration.

Response: We agree with the reviewer that it is not appropriate to refer to the compound as a sex pheromone, but it is correct to use the term pheromone. The bioassays indicate the presence of volatile compounds produced by males that influence conspecific females’ behavioural responses. Therefore, as the action is intraspecific, it can be considered a pheromone. Accordingly, we removed the term sex pheromone but retained the use of pheromone.

  1. Materials and Methods2.1:it is stated that "the individuals used in the experiment were randomly selected from adults, with no control over age, reproductive status, or mating history." However, the production and release of sex pheromones are often influenced by these factors. It is recommended that in key bioassays (such as the attraction of females to male volatiles), clearly unmated individuals of similar age should be used, and their reproductive status should be reported.

Response: We agree with the reviewer that for pheromone studies it is important to have a control of age, and for some species, mainly ones that mate only once, it is important to work with unmated individuals. However, although we were not able to control these parameters in this study, the insects showed differential responses to male. Indicating the possibility of potential pheromone. In the discussion we mentioned on lines 295-302 the importance to evaluate the emission and behavioural response using unmated insects.

“Male of leafhoppers mate multiple times throughout their life [9]. Therefore, it is reasonable to expect that mated individuals may continue producing and emitting sex pheromones. The results obtained in this study appear to support this hypothesis, as part of the insects used were mated, and males were still able to attract females in olfactometer bioassays, even though females’ mate only once in their lives [9]. Further research should be conducted using virgin males and females to assess whether virgin individuals exhibit stronger behavioral responses in bioassays and whether they produce higher levels of sex pheromones.”

  1. Materials and Methods2.3: For the DM treatment group (without food), volatile compounds were only collected for 24 hours due to high mortality, whereas other groups (e.g., DM-Male, DM-Female) had longer collection durations (eluted every 24 hours, with 72-hour accumulation on weekends). Hunger stress may significantly alter the volatile emission profiles of insects. It is necessary to emphasize this limitation in the discussion, or design experiments to ensure that insects in all treatment groups are in a similar physiological state.

Response: We collected male and female volatiles without food to minimize the influence of plant-derived volatiles, which could mask minor insect-emitted components. We are aware that, under starvation, insects may not release pheromones and other semiochemicals; therefore, we performed volatile collections under different physiological conditions. In the revised manuscript, we added a paragraph in the Discussion remarking that the absence of food can influence pheromone production.  Lines 276-294.

“In both conditions used to collect volatiles from males and females, the main compounds identified were aldehydes, monoterpenes, and linear hydrocarbons. No sex-specific compounds were observed between the volatile profiles of males and females collected without food and those of males and females collected with food. In samples with and without food we mainly found compounds related to maize plants emissions, such as cyclosativene and DMNT. The presence of these compounds, even in the absence of maize plants, may be related to honeydew excretion by the leafhoppers, which could not be controlled under our methodology. In Lycorma delicaluta (White) (Fulgoridae), honeydew volatiles released by males can attract conspecific males [22], therefore, this may represent a form of chemical communication in Auchenorrhyncha insects.

The absence of food can influence pheromone production, as demonstrated by [23]. The stink bug Euschistus heros (Fabricius) ceases sex pheromone production after remaining more than 24 hours without food, before this period males of E. heros remain producing the sex-pheromone. In the present study, we were not able to identify sex pheromone compounds, and therefore cannot determine precisely whether the absence of food affects or not pheromone emission. All insects used in the experiments were fed until the start of the assays, remained without food for only 24 hours, thus we hypothesized that at least during the first’s hours of the aeration the insects had physiological conditions to produce and emit semiochemicals.”

  1. Materials and Methods2.6: It is mentioned that "individuals that did not make a choice were excluded until 30 valid bioassays were obtained". It is necessary to report the total number of tested individuals and the proportion of excluded (non-responsive) ones, so as to evaluate the representativeness of the experimental results and potential biases.
  2. Response: We have added this information to the figure. On the right side, next to each bar, the total number of insects responsive and the total number of insects tested are presented.
  3. It is recommended that in Figures 1 and 2, the results of statistically non-significant tests be uniformly labeled "ns".

Response: Done. We chose to remove the “ns” acronyms and keep only the ** to indicate significant results, in order to make the figures clearer.

  1. 3.1 Results:It is described that males avoid the odor of "non-acclimated" females, but show no preference for the odor of "acclimated" females. Similarly, it is necessary to clarify what "non-acclimated" and "acclimated" specifically refer to (the state of odor-source insects? Or the state of test insects?).

Response: We explain in the methodology lines: 180-185. “Because the odor source insects in the glass container were highly mobile, bioassays were repeated using acclimated individuals. Acclimation was achieved by keeping the insects intended as odor sources inside the container for two hours. After this period, the odor sources were connected to the olfactometer, resulting in clearer response behaviors from the test insects toward the odors being evaluated.”

  1. 3.3 Results / 4 Discussion:GC-MS results showed that the volatile compounds of both sexes are "similar" and "no qualitative differences between the two sexes were observed". However, bioassays revealed that females are specifically attracted to male volatiles (and not attracted to female volatiles), while males specifically avoid volatiles from stressed females. This strongly suggests the existence of sex-specific and state-specific compounds, which may be present at concentrations below the detection limit of GC-MS or subject to co-elution. This contradiction needs to be explored more thoroughly in the discussion.

Response:  The no detection of specific male volatiles in the chemical analysis can be related to the detection limits of the equipment used, the insects can produce the compounds in tiny amount. We emphasize this possibility on lines 269-275.

“The chemical analysis of volatile samples containing the volatiles emitted by D. maidis did not identify male-specific compounds that could explain this attraction. It is possible that the male specific compounds are produced in very tiny amounts, below the detection limits of the equipment used. Nevertheless, the observed females to odor of live males and to male aeration extract support the hypothesize that males produce volatiles working as a sex pheromone.”

  1. It is mentioned that there are indications of sex pheromones in Philaenus spumariusand that males of Lycorma delicatula (Fulgoridae) are attracted to female extracts. The discussion should conduct a more in-depth comparison of the similarities and differences between D. maidis in this study (where females are attracted to males) and these known cases (typically males being attracted to females), and explore their significance in the chemical ecology of Hemiptera.

Response: We added a new paragraph to the discussion section on lines 310-318.

  “In addition, some studies with D. maidis in maize fields using sticky cards collected more males, which was attributed to males being caught in traps while searching for mates and attempting copulation [30,31] even this cannot be confirmed only by sticky cards. In another study the escape behavior of D. maidis was not affected by sex [32], indicating that the species exhibits variation in different aspects and warrants further investigation to better understand the searching behavior of males and females over both short and long distances. Our results may reflect short-distance attraction mediated by pheromone, since the distance from the odor source to the olfactometer was less than 50 cm.”

Reviewer 2 Report

Comments and Suggestions for Authors

I believe the manuscript titled " Males of Dalbulus maidis attract females through volatile compounds with potential pheromone function: a tool for pest management" lacks the necessary scientific rigor for publication in the journal Insects.

The most significant issue is the absence of crucial olfactometer bioassays needed to conclusively determine whether D. maidis emits volatile compounds that elicit a response in conspecifics.

For example, the study only compared the response of males and females to odors from males or females versus a control (air). A similar comparison was made with the eluted volatile compounds versus a solvent control. This limited experimental design prevents the authors from making definitive claims.

A more scientifically rigorous approach to olfactometer bioassays should include the following: 1 It is essential to compare the response of males and females to air vs. air. This type of control is critical for minimizing potential errors caused by environmental factors like light, air currents, or other factors that might influence the specimens' preference for one arm of the Y-tube.

2.-  It was necessary to directly compare the response of both sexes of D. maidis to odors from females versus males. This fundamental comparison, which was not performed, is crucial for any study on sexual chemical communication.

3.- It would have been excellent to compare the response of males and females to the odors of males on a food source vs. females on a food source. It is possible that one of the sexes could emit a type of pheromone during the feeding process. In the same way, it was necessary to evaluate the response of both sexes of D. maidis to the eluted compounds.

5.- In addition to the above, if males were emitting a sexual pheromone, it was necessary to determine whether its release was tied to the feeding process. Therefore, a comparison of the D. maidis response to males without food and males with food was also required.

Further observations on these and other points are provided in the attached text.

Author Response

Reviewer 2

I believe the manuscript titled " Males of Dalbulus maidis attract females through volatile compounds with potential pheromone function: a tool for pest management" lacks the necessary scientific rigor for publication in the journal Insects.

The most significant issue is the absence of crucial olfactometer bioassays needed to conclusively determine whether D. maidis emits volatile compounds that elicit a response in conspecifics.

For example, the study only compared the response of males and females to odors from males or females versus a control (air). A similar comparison was made with the eluted volatile compounds versus a solvent control. This limited experimental design prevents the authors from making definitive claims.

A more scientifically rigorous approach to olfactometer bioassays should include the following:

1 It is essential to compare the response of males and females to air vs. air.  This type of control is critical for minimizing potential errors caused by environmental factors like light, air currents, or other factors that might influence the specimens' preference for one arm of the Y-tube.

Response: We absolutely agree. We conducted bioassays using air vs. air as the control, but initially did not include these results in the manuscript, as they did not influence the discussion. However, we recognize the importance of these bioassays, and as suggested by the reviewer, we have now incorporated them into the manuscript.

2.- It was necessary to directly compare the response of both sexes of D. maidis to odors from females versus males. This fundamental comparison, which was not performed, is crucial for any study on sexual chemical communication.

The experimental design as described appears to have a critical limitation. It is not clear why the authors did not compare the behavioral response of males and females of Dalbulus maidis to odors from females versus males. The current comparison of the response of both sexes to live individuals versus air is insufficient. To provide a conclusive argument, it is necessary to test all possible combinations and rule out confounding factors. This methodological gap significantly affects the conclusions of the study.

Response: We understand the suggestion; however, we believe that this additional experiment is not necessary. In the proposed setup, one arm of the olfactometer would contain live males and the other live females. Our previous tests already show that females are neither attracted to nor repelled by other females (female vs. air), but are attracted to males (male vs. air). Similarly, males are neither attracted to nor repelled by either males or female’s vs air. Biologically, this indicates that only males emit volatiles that are attractive to females. In nature, insects are constantly contrasting odours, so we acknowledge that volatiles released by females may influence female responses when tested against males. Nevertheless, these assays would not provide evidence of which sex produces specific compounds that attract the opposite sex, which was clearly demonstrated in our bioassays.

3.- It would have been excellent to compare the response of males and females to the odors of males on a food source vs. females on a food source. It is possible that one of the sexes could emit a type of pheromone during the feeding process. In the same way, it was necessary to evaluate the response of both sexes of D. maidis to the eluted compounds. 5.- In addition to the above, if males were emitting a sexual pheromone, it was necessary to determine whether its release was tied to the feeding process. Therefore, a comparison of the D. maidis response to males without food and males with food was also required.

Response: We thank the reviewer for this observation. Could be interesting to compare males on a food source versus males without food, or males on a food source versus the food source alone, to see if more volatiles are emitted in the presence of food. However, the food source, maize, is the same for females and also serves as an oviposition site. In this way, the experiment would be too complex to explain and could introduce bias. In unpublished data, we conducted bioassays with maize with leafhoppers (both sexes) versus undamaged maize, and females preferred the undamaged maize, probably due to herbivore-induced plant volatiles serving as cues for an optimal place to oviposit and feed. Therefore, conducting this experiment would also require exploring how to balance these herbivore-induced plant volatiles. We can conduct these experiments for a next paper, involving the influence of food in pheromone production.

Further observations on these and other points are provided in the attached text.

Text Considerations

Change to italics

Response: Done

Please provide a clear justification for the use of this specific genotype. Why were common maize varieties not considered for this study?

Response: This genotype was used, because it has been used in our studies, we know well the chemical profile of defensive compounds and volatiles of this genotype for other herbivores (Michereff et al., 2018, 2021, 2022). In addition, this genotype has supported the insect colony for nearly three years, indicating that it does not interfere with development of the insect.

MICHEREFF, MIRIAN F. F. 2022 Neotropical maize genotypes with different levels of benzoxazinoids affect fall armyworm development. PHYSIOLOGICAL ENTOMOLOGY, v. 1, p. 1-10, 2022

MICHEREFF, MIRIAN F. F. Priming of indirect defence responses in maize is shown to be genotype-specific. Arthropod-Plant Interactions, v. 1, p. 1-10, 2021

MICHEREFF, MIRIAN F. F.  Variability in herbivore-induced defence signalling across different maize genotypes impacts significantly on natural enemy foraging behaviour. JOURNAL OF PEST SCIENCE, v. 1, p. 1-10, 2018

"In this way, we added: 'Maize plants of the Synthetic Spodoptera (SS) genotype were used in the experiments, since their herbivore-induced plant volatiles are well known [19],' as this paper was conducted directly with this genotype and D. maidis."

19 - Sanches, M.S.; Michereff, M.F.F.; Borges, M.; Laumann, R.A.; Oliveira, C.M.; Frizzas, M.R.; Blassioli-Moraes, M.C. How much, how long and when: density, duration and plant stage affect herbivore-induced plant volatiles in maize by the corn leafhopper. J Chem Ecol 2025, 51(3), 53.

Did the corn plants have a substrate? Was the entire plant used or only a part of it? I suggest you explain this section in more detail.

Response. Done “The maize plants were placed intact in the system, with aluminium covering the soil to prevent volatiles from originating from the soil or roots”

Please provide a detailed description of the methodology used for sexing the insects. To ensure reproducibility and clarity, it is highly recommended to include a clear image illustrating the morphological differences between males and females. Alternatively, the authors could cite relevant literature that describes the sexing procedure. The former option is strongly encouraged as it would significantly enhance the manuscript.

Response: Done, we made a Supplementary Material for that.

Can a V4-stage corn plant fit completely inside the glass chamber with the specified dimensions? It may or may not, which is why it is necessary to specify how the plants were introduced into the chambers.

Response: It was a mistake in the previous version of the text. In this specific experiment, we used maize plants at the V2 stage (two expanded leaves). Since V4 plants were used in other studies, we inadvertently reported the wrong developmental stage.

Please add the manufacturer, brand, and key characteristics

Response: Done. “… vacuum pump (LGI-DVP-1, ultimate vacuum 200 mbar, pumping speed 60 L/min, LGI Scientific)”

It would be highly beneficial for the reproducibility of the study to include a diagram or photographs of the experimental installation. Such visuals would clearly illustrate the connections and the glass chambers with their assigned treatments. Specifically, a visual depiction of how the V4-stage maize plant was introduced into the chambers is necessary to address the methodological concern regarding its size and placement.

Response: We agree. We include this in Supplementary Material.

Specify which solvent was used for the extraction. The choice of solvent is critical, as it can significantly influence the quantity and profile of the extracted compounds.

Response: Done. “Volatiles trapped on adsorbent filters were eluted every 24 hours with n-hexane”

It is recommended that you add a section or sentence describing the technique used to transfer the Dalbulus maidis specimens from the rearing population to the experimental setup.

Response: Done. “The insects were transferred to the experimental arenas using an insect mouth aspirator [18], from the rearing cages to glass tubes and subsequently to the olfactometers.” The reference provides full details on the rearing and management of the leafhoppers used in the experiments.

The text seems to indicate that the bioassay was replicated to measure the response of males to the odor of 20 live specimens of both sexes. Please provide a clear justification for why this replication was performed exclusively for males and not for females.

Response: We removed the sentence since there is no reason for it to be there. A mistake.

Does this refer to 20 individuals of each sex? If so, please rephrase the sentence for better clarity. Furthermore, it has been previously stated that the total number of individuals was 20

Response: We agree. We rephrased the sentence. “Female and male responses, separately, were tested by contrasting the odor of live 20 males or 20 females versus clean air. Odor sources were generated by placing the insects inside a 40 mL glass container (5 cm x 6 cm) without food.”

I recommend moving the dimensions of the glass container to the previous paragraph, where the Y-tube olfactometer is described. This would keep all relevant setup details in a single section.

Response: Done.

The preliminary bioassays with unacclimated insects were useful for fine-tuning the experimental technique. It is appropriate to mention this process, but the results from these trials do not necessarily need to be included in the main body of the paper's results.

Response:  Yes, these preliminary bioassays with unacclimated insects were mainly useful for fine-tuning the experimental methodology, but we observed a repellence response of males when exposed to the odour of non-acclimated females versus air. Because this response suggests the possible involvement of an alarm pheromone, we considered it important to mention these results in the manuscript.

"One insect".. What was the sex of the insects? Were only females, only males, or both used? Please specify.

Clarify the exact location of the insect release point. Was a connector or base used just before the Y-tube entrance? It's important to specify if the insects were released directly into the tube's entrance or from another area.

 Response: Done. “One insect (male or female) was released in the entrance of the olfactometer…”

I do not fully understand this part of the methodology. The insects whose response was evaluated—did they remain inside the Y-tube or in another part of the olfactometer? Please explain this in more detail.

Response: We wanted to show that the acclimation was conducted with the 20 insects inside the glass container (odor source). At the same time, we also wanted to indicate that each insect tested was acclimated in the olfactometer before the 10-minute bioassays, waiting for the fast movements and jumps to stop, which varied for each insect but never exceeded one minute. To avoid confusion, we removed the second part, as it is not essential for the experiment reported in this paper.

I suggest restructuring this paragraph and improving its flow. (Lines 166-176)

Response: Done.

The methodology states that initially only 20 unacclimated females were placed in the glass chambers, with no mention of males. Also, these activities were preliminary tests to improve the bioassay technique, so their results shouldn't be presented as final findings.

Response: The sentence mentioning only females was incorrect, and we have changed it to “insects.” As we explained previously, one part of this experiment could support the release of semiochemicals by D. maidis, and their potential function as alarm signals, since the insects were stressed.

What is the reason for the presence of these compounds in the odor extracts of individual females and males? These compounds are generally common in plants

Response: We also found this result unexpected, which is why we conducted additional male and female volatile collections without maize plants. Even so, these compounds constantly detected in all samples from both sexes. We checked for contamination in the chambers and adsorbents, but nothing was detected. Therefore, we suggest that honeydew excretions from these leafhoppers, which we could not control during the aeration, may explain the presence of these volatiles. We added this information on discussion “The presence of these compounds, even in the absence of maize plants, may be related to honeydew excretion by the leafhoppers, which could not be controlled under our methodology. In Lycorma delicaluta (White) (Fulgoridae), honeydew volatiles released by males can attract conspecific males [22], therefore, this may represent a form of chemical communication in Auchenorrhyncha insects.”

The conclusions drawn from this study are not sufficiently supported by the presented data.

Response: We do not agree with this comment. We made it clear that this is a potential pheromone behavior, as only females were attracted in one of the sexes across three different bioassays: non-acclimated males, acclimated males, and male aeration extract (90 bioassays in total).

The conclusion related to chemical communication and insect behavior is overly speculative. This study does not definitively demonstrate this claim. At most, the findings only suggest the presence of volatile compounds involved in the behavior of D. maidis.

Response: Our study demonstrates that males release volatile compounds that attract conspecific females. Since the observed chemical communication is intraspecific, we can state that male-emitted pheromones are involved in this process. As males were the only odour source in the chambers, the volatiles detected must originate from them. Our results consistently showed that females were attracted to males in 90 bioassays. Females’ response were tested in different days to different treatments containing the male odour: a) to odour emitted from non-acclimated male vs air, b) to male vs air, and c) male aeration extract vs air). In contrast, no attraction or repellence was observed in female–female, male–female, or male–male tests. This provides clear evidence of sexual behavior and communication between the sexes via volatiles. It is worth emphasizing that only two studies exist on Auchenorrhyncha to date, highlighting that studying chemical communication in these organisms is a challenging task. Our work conducted a series of bioassays and present results, as previous studies suggested chemical communication among D. maidis males, and our findings now confirm this, with olfactometers providing a consistent methodology for this type of experiment.

Reviewer 3 Report

Comments and Suggestions for Authors

Review of the manuscript: “Males of Dalbulus maidis attract females through volatile compounds with potential pheromone function: a tool for pest management”.

The results presented in the manuscript are the part of the research on insect sex pheromones which help to monitor the pest population. Pheromones seem to be one of the most successful applications in the pest management among others because of their high efficacy at low population density and they do not affect on natural enemies. That is why research on sex pheromones especially for new species are great of value.

The reviewed manuscript is well written and structed as well as contains information of interest to readers. Appropriate study methods were used and the results were described correctly. Discussion is interpreted in detail. I have only minor comments:

- the subsection 2.1.: please describe the raring of Dalbulus maidis, because the reference 18 cited here is not available in open access mode (at least I couldn't find it) so it is not known how this procedure was carried out exactly;

- the subsection 2.3.: please describe how long the entire experiment persist. It states that the insects and plants were replaced weekly, but how long did this take in full? This is quite important information.

Author Response

Reviewer 3.

Review of the manuscript: “Males of Dalbulus maidis attract females through volatile compounds with potential pheromone function: a tool for pest management”.

The results presented in the manuscript are the part of the research on insect sex pheromones which help to monitor the pest population. Pheromones seem to be one of the most successful applications in the pest management among others because of their high efficacy at low population density and they do not affect on natural enemies. That is why research on sex pheromones especially for new species are great of value.

The reviewed manuscript is well written and structed as well as contains information of interest to readers. Appropriate study methods were used and the results were described correctly. Discussion is interpreted in detail. I have only minor comments:

- the subsection 2.1.: please describe the raring of Dalbulus maidis, because the reference 18 cited here is not available in open access mode (at least I couldn't find it) so it is not known how this procedure was carried out exactly;

Response: Done. We changed the reference because we noticed that the English version of this book is not available online.

- the subsection 2.3.: please describe how long the entire experiment persist. It states that the insects and plants were replaced weekly, but how long did this take in full? This is quite important information.

Response: Done. “Volatile organic compounds (VOCs) emitted by D. maidis were collected using dynamic headspace aeration systems across two experimental sets that last for 2 weeks each.”

Round 2

Reviewer 2 Report

Comments and Suggestions for Authors

Overall, the manuscript has improved compared to the previous version. The inclusion of images is helpful in illustrating how to distinguish between sexes of D. maidis, as well as in showing the setup of the equipment used to capture volatiles from the different treatments.

The authors provide a reasonable response to my previous comments regarding the absence of olfactometer bioassays. However, in my view, it was necessary to compare the behavioral responses of male and female D. maidis to odors emitted by conspecific males versus females (either live specimens or extracts). In contrast, the authors argue that such bioassays were unnecessary, as they had prior evidence indicating that females only respond to male odors.

Based on my experience, olfactometer experiments can be influenced by various factors that may lead to false positives, and it is always advisable to minimize potential sources of bias as much as possible. Indeed, it is very likely that, when comparing male versus female odors, only females would respond to males; however, we cannot know this with certainty. The chemical ecology of insects is inherently complex, and to properly understand these interactions, all possible alternatives must be carefully ruled out. Furthermore, the authors did not identify specific male-emitted volatiles that could explain the observed female attraction.

It is possible that our differences arise from methodological approaches to reaching the expected results; such discrepancies are common and even represent a healthy part of the scientific process. Nevertheless, I maintain that the authors cannot conclusively claim that their experiments demonstrate the role of chemical communication in the sexual behavior of D. maidis. I emphasize that their results merely suggest the involvement of volatiles in the species’ sexual behavior, but they do not provide definitive confirmation.

Finally, I would like the authors to address one last question before I reach a decision: How can you be certain that female attraction to males is due to chemical compounds rather than to the emission of specific sounds produced by the males?

Round 3

Reviewer 2 Report

Comments and Suggestions for Authors

Well done. We can all now agree that the work and the writing have significantly improved compared to the first version. Congratulations to the authors.